# Deep learning classification of lipid droplets in quantitative phase images

Luke Sheneman[1]*, Gregory Stephanopoulos[2], Andreas E. Vasdekis[3]*

**1** Northwest Knowledge Network, University of Idaho, Moscow, Idaho, United States of America,
**2** Department of Chemical Engineering, Massachusetts Institute of Technology, Cambridge, Massachusetts, United States of America, **3** Department of Physics, University of Idaho, Moscow, Idaho, United States of America

* sheneman@uidaho.edu (LS); andreasv@uidaho.edu (AEV)

## Abstract

We report the application of supervised machine learning to the automated classification of lipid droplets in label-free, quantitative-phase images. By comparing various machine learning methods commonly used in biomedical imaging and remote sensing, we found convolutional neural networks to outperform others, both quantitatively and qualitatively. We describe our imaging approach, all implemented machine learning methods, and their performance with respect to computational efficiency, required training resources, and relative method performance measured across multiple metrics. Overall, our results indicate that quantitative-phase imaging coupled to machine learning enables accurate lipid droplet classification in single living cells. As such, the present paradigm presents an excellent alternative of the more common fluorescent and Raman imaging modalities by enabling label-free, ultra-low phototoxicity, and deeper insight into the thermodynamics of metabolism of single cells.

## Introduction

Quantitative-phase imaging (QPI) of biological systems has met with significant success in recent years, both in fundamental and biomedical investigations [1–5]. In QPI, an image is formed by quantifying the optical path length (or optical phase delay) difference between the specimen and its background without any fluorescent staining [6–8]. This type of imaging confers unique information about cellular physiology and metabolism, including cellular dry-mass and dry-density that are not attainable via conventional, volumetric, bioimaging methods [9]. Further, by detecting optical phase rather than intensity (Fig 1a), QPI enables high-contrast cell imaging with minimal background heterogeneity, thus enabling the contour localization of individual cells (i.e., performing cell segmentation) without any computationally intensive approaches [10,11]. More recently, QPI has been interfaced with deep learning methods in various applications ranging from image reconstruction to object recognition [12–18]. In the context of object recognition, applications such as screening histological targets [19], colorectal cancer [20] and hematological disorders [21,22], as well as the detection of nuclei in structural biology applications [23] have been demonstrated.

**Data Availability Statement:** All source code related to this manuscript is available via a public Github repository at: https://github.com/sheneman/deep_lipid. All project data is now publicly available. This specifically includes: QPI-acquired

image data used for machine learning training, validation, and testing. These data are publicly available via the University of Idaho's institutional FAIR data repository and may be accessed here: https://doi.org/10.7923/3d0d-yb04.

**Funding:** The work described in this manuscript was funded by a U.S. Department of Energy (DoE) grant (DE-SC0019249) entitled, "Multi-Modal Imager of Metabolome and Enzyme Dynamics for Co-Optimizing Yields and Titers in Biofuel Producing Microorganisms". Two manuscript authors: AEV (Project PI) and LJS (Project Senior Personnel) received funding under this DoE award. URL of funder: www.doe.gov The funder (DoE, Grant #DE-SC0019249) provided funding for the study design, data collection/analysis, and preparation of the manuscript. However, the funder was not an active hands-on participant in any of these activities.

**Competing interests:** The authors have declared that no competing interests exist.

Here, we report the application of deep learning in recognizing and localizing lipid droplets (LDs) directly in quantitative-phase images without any staining. LDs are an important cytosolic organelle where all eukaryotes (and some bacteria) store neutral lipids in the form of triacylglycerols, steryl and retinyl esters [24–26]. Catabolic utilization of these compounds provides cells with membrane building blocks and energy in the form of ATP, indicating an important role of LDs in energy homeostasis and metabolism [27,28]. Recent evidence has expanded the role of LDs as a key trafficking node of proteins, transcription factors and chromatin components [29–33]. The implications of LDs in disease have also been better understood in recent years, primarily in the context of infection by intracellular pathogens [34,35] and viruses [36,37], as well as fatty-liver disease [38] and various forms of cancer [39]. Further, LDs have found tremendous applications as a sustainable source of biodiesel production [40–42].

The ability to visualize LDs at the single-cell level has greatly advanced our understanding of the key roles that LDs play in metabolism, disease, and biotechnology, as well as has unmasked the underlying effects of cellular noise [43–45]. While fluorescence and Raman microscopy have been the most common imaging methods in such investigations [46–51], LD localization by QPI has also been recently demonstrated [52,53]. The advantages of QPI over other imaging methods pertain primarily to label-free imaging with ultra-low phototoxicity (typically at $\mu W/cm^2$ illumination densities) that can confer key thermodynamic parameters, such as the ensemble number-density of molecular assemblies. However, due to the relatively low refractive index contrast between LDs and other organelles, LD localization in single-living cells by QPI exhibits insufficient discriminatory power and, thus, low specificity (Fig 1b). As such, QPI is not compatible with automated high-throughput image processing in LD

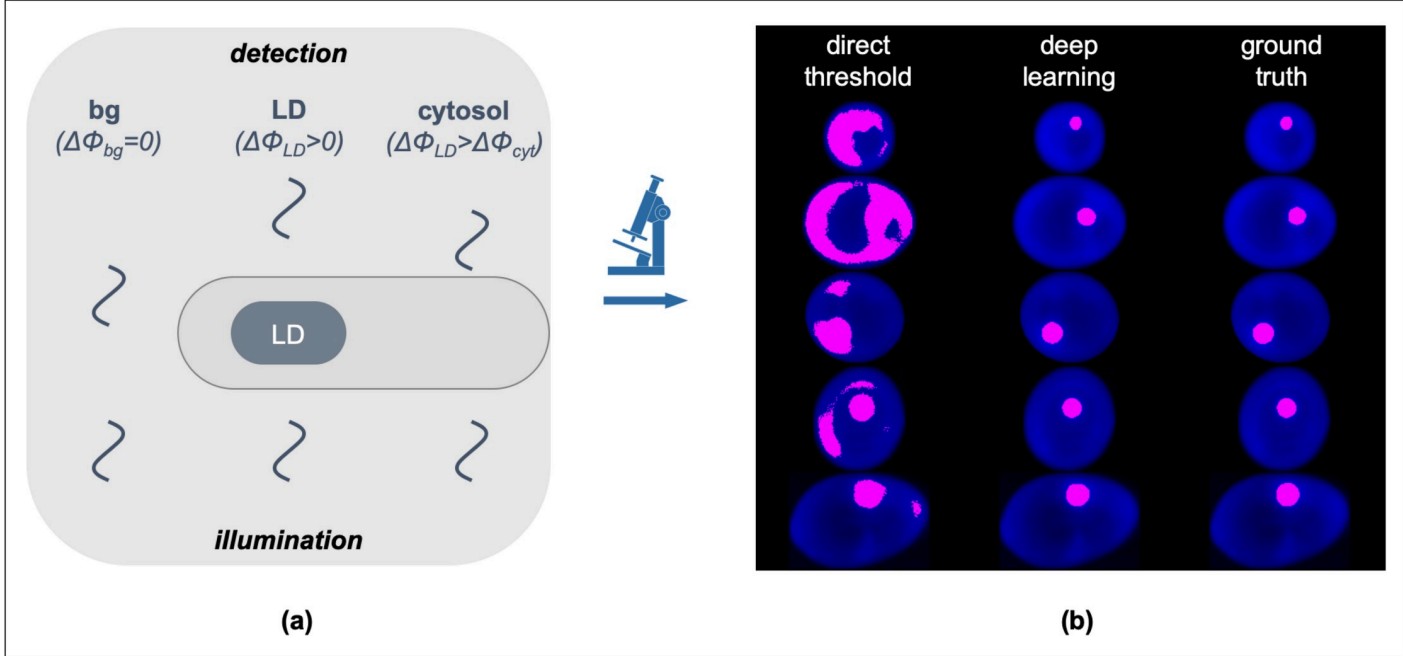

**Fig 1.** **(a)** Cartoon diagram of the QPI image formation through the detection of the optical phase-delay of light transmitted through the cell cytosol ($\Delta\Phi_{cyt}$) and cytosolic LDs ($\Delta\Phi_{LDs} > \Delta\Phi_{cyt}$ due to the innate refractive index differences) at a constant background ($\Delta\Phi_{bg} = 0$). **(b)** Representative Yarrowia lipolytica QPI images (in blue) overlaid with binary masks (magenta) acquired by direct thresholding (left column) and deep learning (middle column). The decreased discriminatory power of direct thresholding and the increased precision of deep learning become evident upon comparison with the ground truth (right column).

localization, with the exception of coupled deconvolution- and correlation-based post-processing schemes, albeit at increased computational resource requirements and error-rates [53].

To overcome these shortcomings and enable fully automated LD recognition in QPI (Fig 1b), we explored six distinct supervised machine learning methods. Using *Yarrowia lipolytica*, a tractable oleaginous yeast [54–56], we explored ensemble decision tree models including random forest [57,58] and gradient boosting [59,60], multilayer perceptron [61,62], support vector machines [63–66], linear discriminant analysis [67–70], and deep learning convolutional neural networks (CNN) [71–74]. We found that CNNs generally outperform other methods across a number of quantitative metrics, including accuracy, balanced accuracy [75], precision and recall [76,77], Jaccard distance [78–80], and the F1 Score or Sørensen-Dice coefficient (i.e., "Dice scores" [81–84]). In the following sections, we describe the generation of binary training images, detail each machine learning method, and compare their performance both quantitatively and qualitatively.

## Methods

### Constructing the curated image library

We started by labeling a large image library of individual *Y. lipolytica* cells that we have previously reported [53]. The library consists of two *Y. lipolytica* strains, Po1g and MTYL038 [54], with the former being auxotrophic for leucine (Leu⁻) and yielding 1.5% greater lipid content per weight at identical growth conditions (i.e., 100 hrs in a carbon-to-nitrogen ratio (C/N) equal to 150) [53]. The two strains displayed similar images at these conditions and were not labeled specifically during training and classification. To additionally include images with both low (smaller LDs) and high (larger LDs) lipid content, MTYL038 images were collected at different timepoints (28 hr, 52 hr, 76 hr, and 124 hr) in the same medium (C/N = 150). In this context, cells exhibiting both low (early time points) and high (late time points) lipid content were attained [53]. Similarly, the different time points for MTYL038 were not labelled specifically during training and classification. We found that our curated image library is sufficiently large to train robust classifiers without the need for data augmentation of the acquired data. As detailed in [53], cells were transferred from rich YPD medium (20 g/L Bacto Peptone from BD, 10 g/L yeast extract from Alfa Aesar, 20 g/L glucose from Fisher) to the defined YSM medium (1.7 g/L yeast nitrogen base without amino acids and without ammonium sulfate from BD Difco) 0.69 g/L complete supplement mixture without Leucine from Sunrise Science Products, 1.1 g/L ammonium sulfate from Fisher and 75 g/L glucose from Fisher) at 50 × dilution (dilutions yielding starting ODs at 0.01) and centrifugation (490 × g) and washing in YSM three times. For the Po1g strain, 0.1 g/L Leucine was added in the YSM medium. All growth experiments were performed in a shaking incubator at a temperature of 29 °C.

To perform quantitative phase imaging, we placed 2 μL from a growing culture between two coverslips and pressed gently to minimize the distance between the two coverslips prior to imaging. We acquired all images using a quantitative-phase imaging system (Phi Optics) that relies on the spatial light interference microscopy (SLIM) modality. In SLIM, images are formed with the aid of a liquid crystal phase modulator that shifts the optical phase of the light wavefront scattered by the specimen with respect to un-scattered light [85]. This system was coupled to an inverted microscope (DMi8, Leica) equipped with an automated *xyz* stage, a 100× magnification objective (NA 1.3, PH3), and a sCMOS camera (Orca Flash 4.0, Hamamatsu). 3D images were acquired by scanning the objective along the imaging path (*z*-axis) with a step of 400 nm and the stage in the *xy* plane. Due to the lower axial resolution of our imaging set-up than the lateral one, we compressed the acquired 3D images into 2D via the maximum

projection method. Using the 2D images, we detected the cell contour by direct optical-phase thresholding and no further pre-processing [10].

To localize LDs and generate a binary training data-set, we first 3D deconvoluted two phase-modulated intensity images [$I_{\pi/2}(x,y,z)$ and $I_\pi(x,y,z)$] using the AutoQuant X3 procedure (MetaMorph), and subsequently cross-correlated these two images. As we have previously described, this approach enabled quasi-automated LD segmentation that, however, suffered from increased computational times (indicatively, the deconvolution step required more than 12 h per 2,000 images) and increased error-rates [53]. To minimize errors during training and classification, we further evaluated each single-cell image and excluded subsets with noticeable defects (e.g., specks of dust occluding the image), apoptotic cells, small buds, and cells that spatially overlapped. Our final usable image library consisted of approximately 7,000 32-bit grayscale quantitative phase images of single *Y. lipolytica* cells. With the goals of performing controlled tests while evaluating the effect of training set size on performance and accuracy, we randomly split our image library into a fixed set of 2,000 test images and 5 different training sets ranging between 1,000 images and 5,000 images (Fig 2). These test and training sets were then used in subsequent evaluations, including a k-fold cross-validation [86–88] comparison of all machine learning methods using the stochastically split library of 5,000 training images.

## Non-deep machine learning methods

We implemented a number of traditional machine learning methods including linear discriminant analysis (LDA), support vector machines (SVM), multilayer perceptron neural networks (MLP), random forest (RF) and gradient boosting (XGB) via *XGBoost* [89]. These machine learning methods have demonstrated effectiveness with the semantic segmentation of images across a number of disciplines including bioimage analysis [87,90–96] and remote sensing [97–102]. Notably, classifier models trained with ensemble methods such as random forest and gradient boosting benefit from a relatively high level of interpretability [103–105] as the individual estimators (decision trees) within these models can be understood and interpreted without significant effort. As such, the contribution and importance of individual features can be estimated [106,107] and subsequently used to optimize feature selection [108–110] (S1 File).

The training and classification workflows are the same for all five of our non-deep learning methods (Fig 3, S1 Fig). During training, we read all *Y. lipolytica* images from our training set as a collection of single-band 32-bit grayscale TIFFs. Each image (Fig 3a) is sequentially passed into a central preprocessing routine which first converts the image to a normalized 8-bit internal representation to improve memory and computational efficiency in exchange for an acceptable loss in classification accuracy. In order to extract features for every pixel of a given image, we then apply a static set of 80 parametrized image filters to the input image. These are common image filters as implemented in OpenCV2 [111], SciPy [112], and the Python Imaging Library (PIL) and include gaussian smoothing, intensity thresholding, Sobel edge detection [113], Laplacian of Gaussian Edge Detection [114], Structure Tensor Eigenvalues [115] and more (see Fig 4 for related examples and S1 File for a full list of the used filters and their parameterization). These filters effectively extract new per-pixel features by computing intensity value changes to any given pixel as a non-linear transformation of the current pixel in the spatial context of the overall image with weighted contributions from neighboring pixels.

Our preprocessing function expands our original 2D single-band image of width *w* and height *h* into a 3D array with a feature depth *k* equal to the number of applied filters (Fig 3b). As it contains intrinsic feature information in its original form, we also retain the original 8-bit

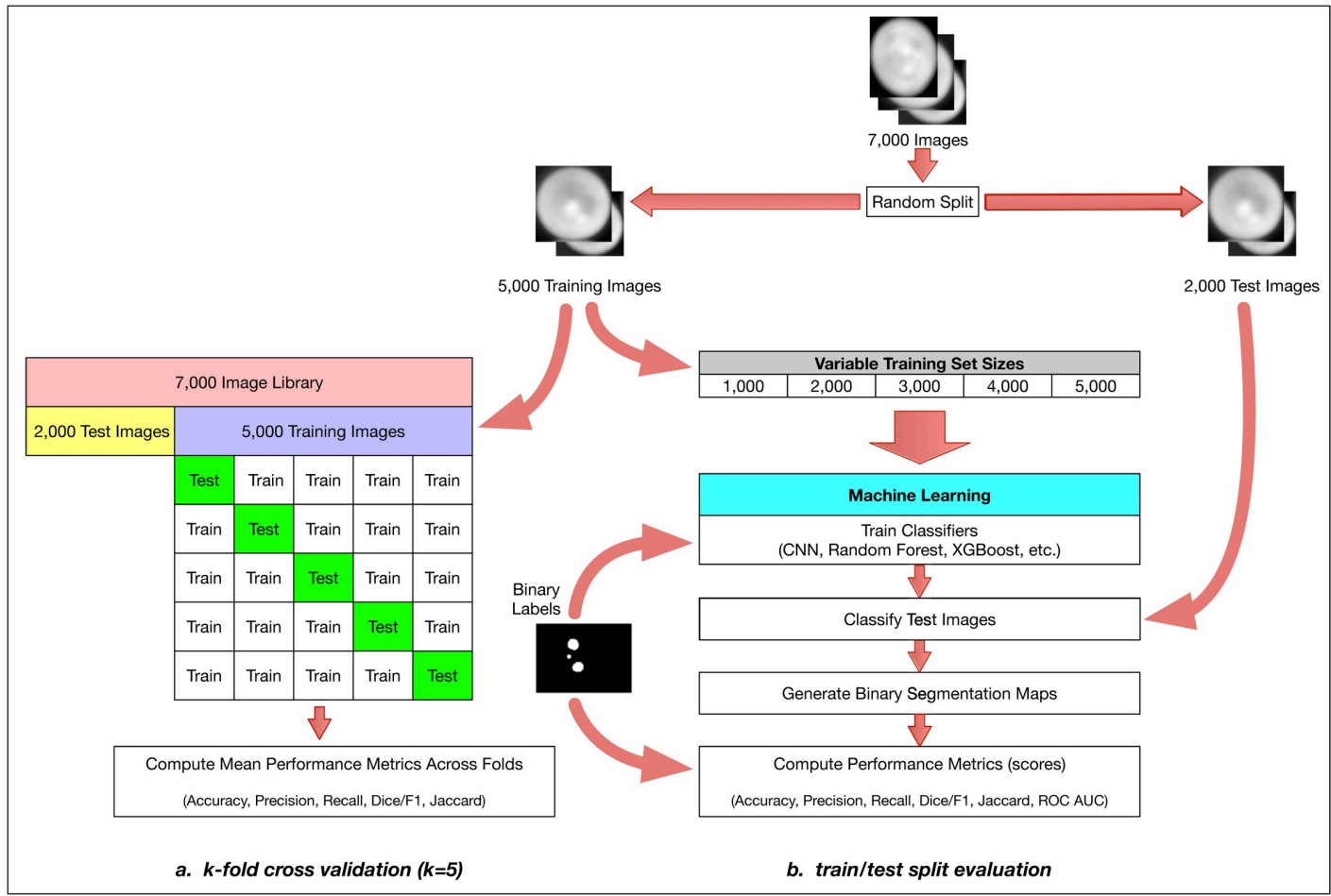

**Fig 2. The overall method design.** We start with approximately 7,000 Y. lipolytica QPI images and perform a train-test split by randomly selecting 5000 images as a training set and 2000 as our test set. We then perform two distinct but related evaluations of multiple machine learning methods: a comprehensive k-fold cross-validation on the training set (**a**) and a limited train-test split evaluation (**b**). For the k-fold cross-validation case, we further split our 5000-image training set with k = 5 without shuffling and iteratively validated against our respective hold-out set across all folds. We report several model effectiveness metrics averaged across all k = 5 folds. For the train-test split evaluation, we randomly subsample training sets of size 1000, 2000, 3000, 4000, and 5000 from our training set. With these training sets, we train six types of machine learning models: convolutional neural networks (CNN), gradient boosting with XGBoost (XGB), random forest (RF), support vector machines (SVM), linear discriminant analysis (LDA), and multilevel perceptron (MLP). For CNNs, we train for 1000 epochs, where an epoch is one full pass of the training data set through the neural network model during training. For the RF and XGB ensemble methods, we use a collection of 100 decision trees/stumps. For the MLP model, we used two hidden layers of 50 and 25 nodes each and for SVM we used a 3-degree polynomial kernel. During training, we use binary labeled data associated with each raw image to train our models. Each method outputs a binary segmentation map which can then be compared to the binary labeled image template to compute model effectiveness metrics (e.g. accuracy, precision, recall, and the Sorensen-Dice coefficient).

image as one of the $k$ feature layers. In total, we preprocess every $i^{th}$ 2D input image of width $w_i$ and height $h_i$ and a depth of $k = 80$ features into a 3D array of dimension ($w_i$, $h_i$, $k = 80$).

We sequentially preprocess every image in our training set of $s$ images in this way and *flatten* our internal 3D representation of the training set data with $k = 80$ extracted per-pixel features into a single long 2D array with $\sum_{i=1}^{s} w_i \times h_i$ rows and $k = 80$ columns (Fig 3c). We separately perform the same flattening operation with our binary training labels into a distinct internal data structure. While this data transformation is computationally expensive, this 2D internal representation of data points and associated extracted features is required for our sci-kit-learn and XGBoost model training functions. We then train our five types of non-deep

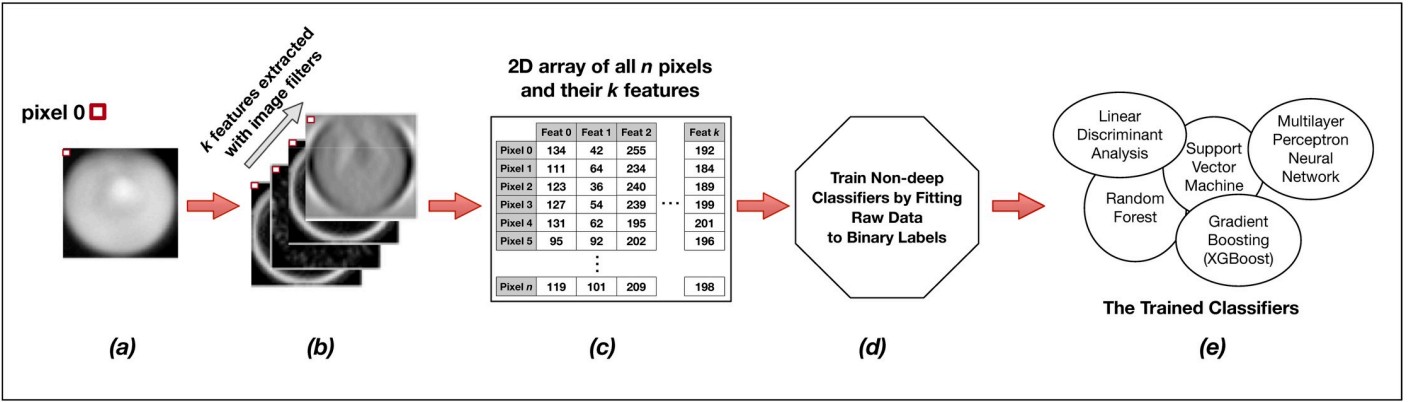

**Fig 3. Training the non-convolutional models.** Each individual raw image (**a**) in the training set of s images is expanded into a set of k layers through the application of parameterized image filters (see Fig 4 below) in order to extract informative features for each pixel (**b**). Individual pixels are then described not by a single intensity value, but by a feature vector of length k. We then reorganize these data for all s images into a long two-dimensional array of n pixels and k = 80 features (**c**) wherein each row represents a single pixel in the training set, and each of k columns represents a specific extracted feature for that pixel. This n x k matrix, along with the n corresponding binary labels for each pixel, is passed as training input to a model fit function (**d**). The output of the training procedure is a set of trained classifier models (**e**).

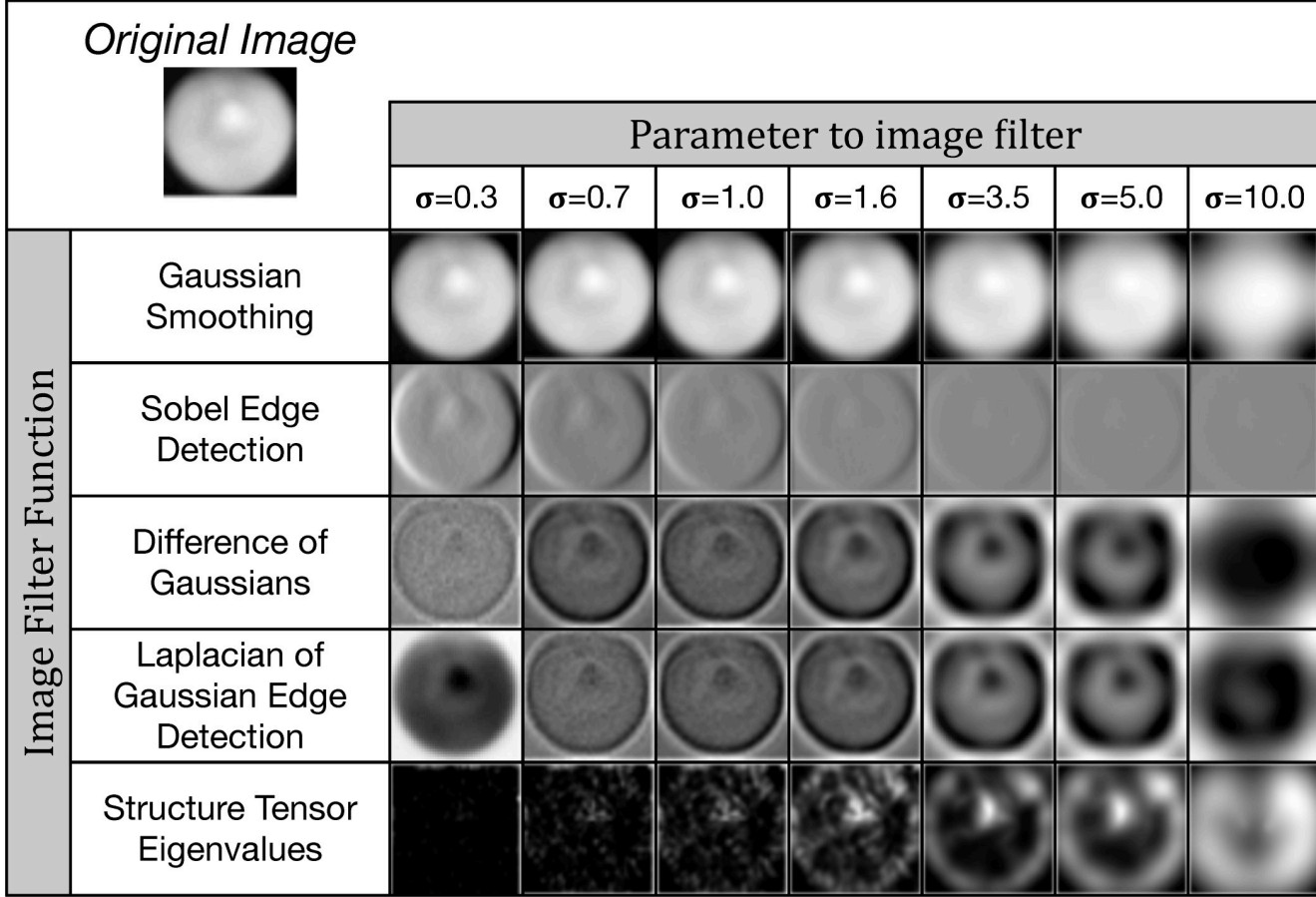

**Fig 4. A subset of the image filters used for feature extraction.** We apply a number of common digital image filters such as Gaussian smoothing and Sobel edge detection to each input image, using multiple σ parameter values. This extracts a total of k = 80 features per pixel. These features are then used for either training or classification in our non-deep (i.e. non-convolutional) machine learning methods.

**Table 1. Hyperparameters for our non-convolutional machine learning methods.**

| Random Forest (RF) | |
|---|---|
| Number of Estimators | 100 |
| Number of Threads | 25 |
| **XGBoost (XGB)** | |
| Number of Estimators | 100 |
| Number of Threads | 25 |
| Max Depth | 4 |
| Subsample | 0.5 |
| **Support Vector Machine (SVM)** | |
| Kernel | Poly |
| Degrees (Poly) | 3 |
| Gamma | "scale" |
| Max Iterations | 1000 |
| Cache Size | 5000 MB |
| **Multilevel Perceptron (MLP)** | |
| Hidden layers (2) | (50,25) |
| Activation Function | ReLU |
| Solver | Adam |
| Max Iterations | 1000 |
| # Iteration w/o improvement | 50 |
| **Linear Discriminant Analysis (LDA)** | |
| Solver | "svd" |

We use parallel computation and multi-threading where possible for improved performance on multi-CPU systems. For XGBoost, we explicitly specify a maximum decision tree depth of 4 and we subsample training data to mitigate overfitting. For SVM, we used a non-linear Polynomial kernel with 3 degrees and a limit of 1000 iterations. For MLP, we used 2 hidden layers of 50 and then 25 nodes each and also limited our optimization to 1,000 iterations with a halt condition of insufficient improvement after 50 iterations.

learning models by fitting them to the training data (Fig 3d) using specific hyperparameters (Table 1).

The model training output is a set of distinct trained classifiers (Fig 3e) that each fit our raw training images to the provided binary training labels. The computational time required to train different machine learning methods varies, but because of the computationally intensive data-flattening step needed prior to model fitting, the computational complexity of our overall training program is at least quadratic at $O(n^2)$ [116], where $n$ is the total number of input pixels across all images in our training set (Fig 5). However, support vector machines as implemented in LibSVM has a known complexity of $O(n^3)$ [117] and is demonstrably harder to parallelize, leading to significantly longer training times than our other evaluated machine learning methods. Trained models are written to disk for later use in classification (semantic segmentation) of new images. These stored machine learning models can often be inspected to report metadata such as relative feature importance (S1 File).

During the classification step, archived trained models are read from disk into memory and new raw images are individually preprocessed using the same collection of image processing filters that were employed for feature extraction prior to model training. Similarly, the preprocessed input image data for each image is flattened prior to use by the set of trained classifiers. Each pixel represented as a $k = 80$ feature vector is passed through the loaded, trained model

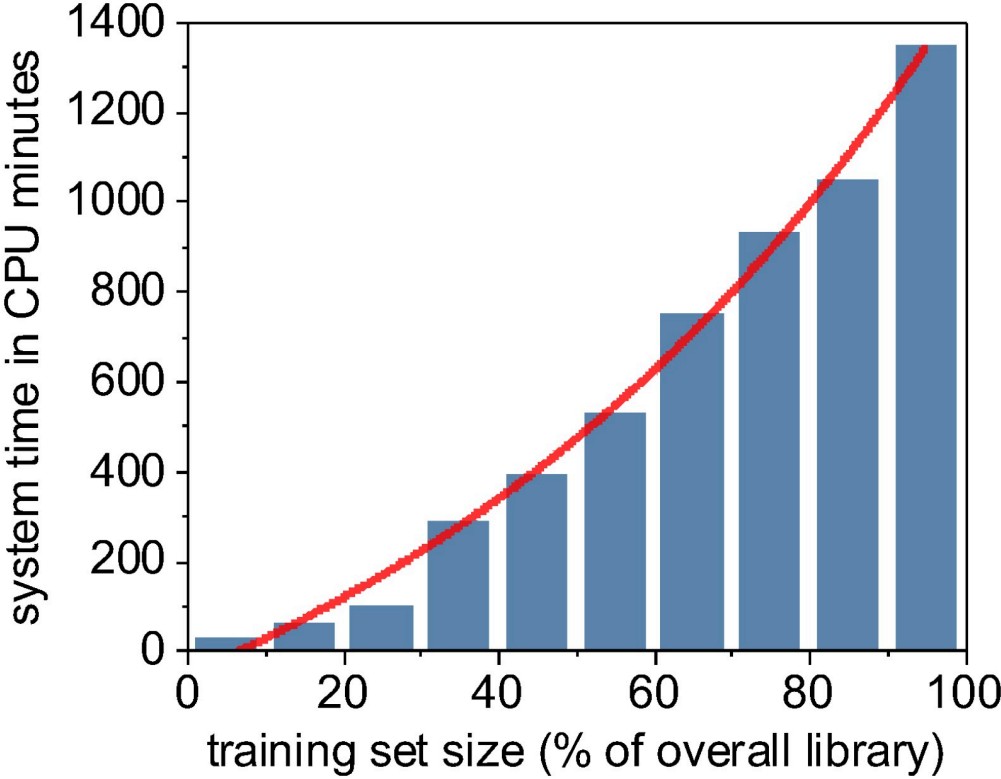

**Fig 5. Computational time for training a random forest model is a superlinear function of training set size.**

in order to classify the pixel as either *lipid* or *non-lipid*. This is repeated for all pixels in a given input image in order to construct an 8-bit binary segmentation map image as output.

### Convolutional neural networks with the U-Net architecture

In recent years, deep learning methods, such as convolutional neural networks (CNNs), have demonstrated remarkable accuracy in difficult computer vision applications such as image classification [118–126]. More recently, powerful CNN architectures have been introduced to perform semantic segmentation of images [127–129]. Originally developed for biomedical image analysis, U-Net is one of the most widely utilized, studied, and cited CNN architectures for image segmentation [130]. We implement U-Net as a deep learning method for performing binary classification of LDs in images of *Y. lipolytica* cells and quantitatively and qualitatively compare model efficacy and computational speed to our results from the non-deep machine learning methods discussed previously.

The U-Net layout is *fully convolutional* and therefore has no fully connected layers (Fig 6). The first layer starts with an original image of a fixed dimensionality, in our case this is a 256x256 single-channel grayscale image. As our *Y. lipolytica* image library contains images of differing widths and heights, we first pad all of our images to be a consistent 256x256 size. This is trivially done by computing the amount of border to evenly add to the top, bottom, and sides of the raw source image to result in a 256x256 image subsequently calling the ImageOps. expand() function within the Pillow (PIL) Python library. The next two convolutional layers are configured to apply 64 image filters using a 3x3 kernel that convolves over the image data from the previous layer. We specifically use padding in our convolutional layers to preserve

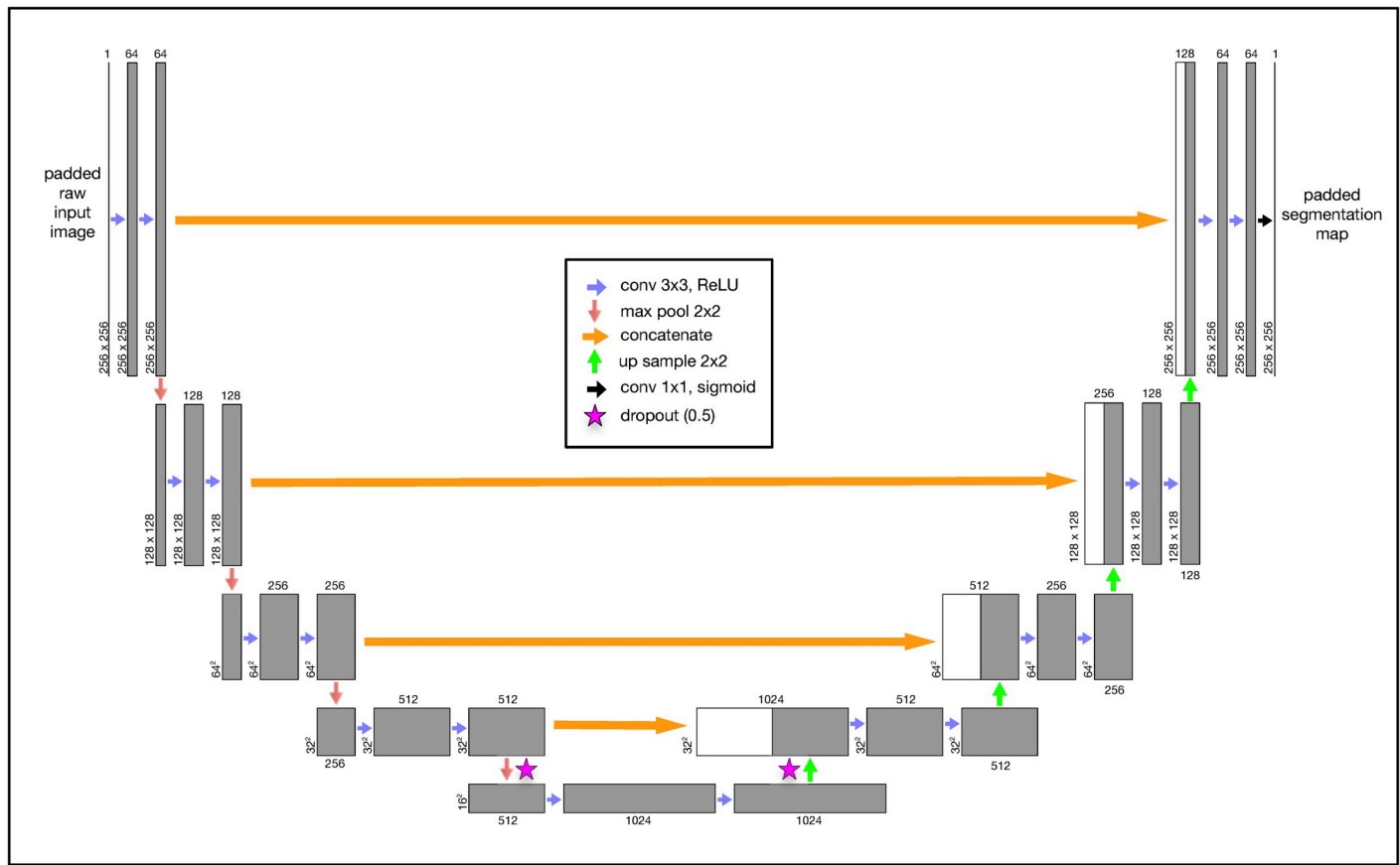

**Fig 6. The U-Net convolutional neural network (CNN) architecture was developed specifically for semantic segmentation.** Our grayscale input image is consistently padded to 256x256. This image is passed through convolutional layers using a Rectified Linear Unit ("ReLU") activation function with a 3x3 kernel. Each convolutional layer specifies padding to prevent gradual image shrinkage. The original image (now transformed into a feature map tensor) passes through a series of convolutional and 2x2 max pool layers until the tensor is finally reduced to 16x16 with a feature depth of 1024. At the lowest levels, we perform a 0.5 dropout to mitigate overfitting. We then iteratively up-sample (2x2) the tensor and perform a second dropout while concatenating it with the earlier tensor of the same dimension at the same level. We perform this same concatenation operation at every up-sample layer. The final convolutional output layer uses a continuous sigmoid activation to approximate a binary classification for each pixel.

the original dimensions of the previous layer. This convolution process of sliding a 3x3 kernel over our image data extracts features in a way that is analogous to our use of image filters in the non-convolutional methods described earlier in this paper. The result of the first three layers is a 256x256x64 *feature map tensor* that represents the original image dimensions, encoded now with 64 extracted features. This tensor is now passed through a 2x2 max pool layer that down-samples the original image width and height by a factor of 2, leading to a 128x128x64 tensor. This is then passed through additional image filters and down-sampled again. This pattern is repeated several times, reducing the tensor's width and height while greatly expanding its feature depth. At the lowest level of the U-Net, we apply dropout layers to randomly sub-sample from the prior convolutional layer in an attempt to mitigate model overfitting to our training data [131]. In total, our U-Net CNN implementation has 31,031,685 trainable parameters.

We then begin to scale up our tensors using 2x2 up-sampling layers which are followed immediately by concatenation with the tensors from prior layers at the same depth. This concatenation step after every up-sampling layer ultimately combines important spatially-correlated elements of the original 2D image with the more deeply transformed feature-rich

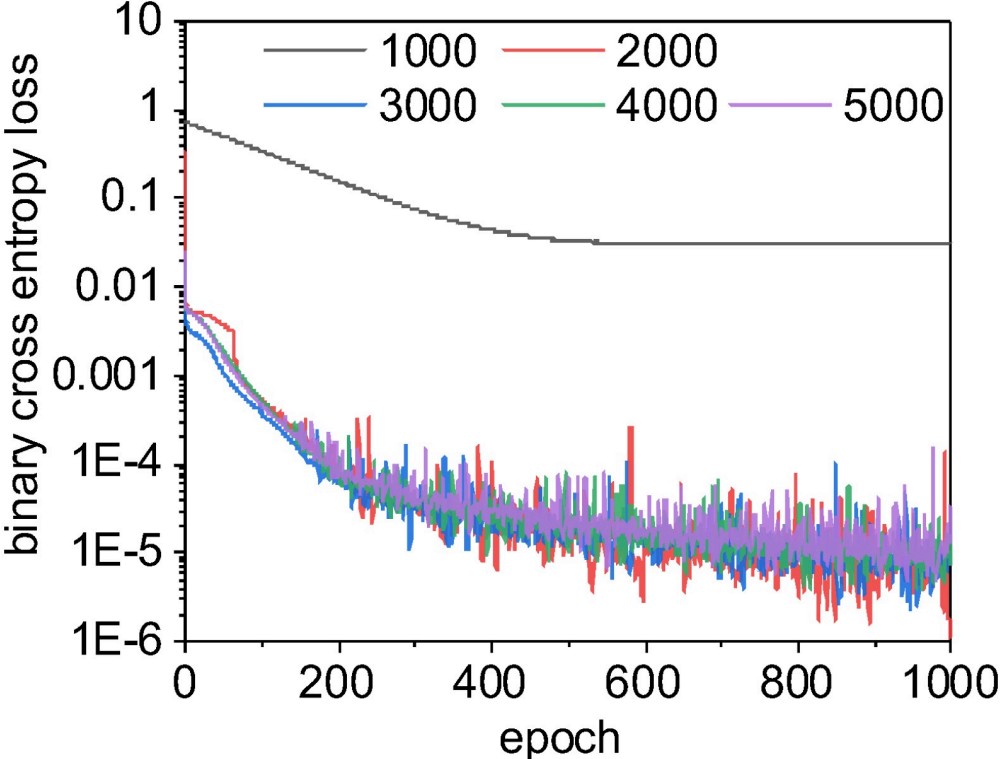

**Fig 7. The U-Net CNN learning curves for different sizes of training sets.** In neural networks, a learning curve is the rate of model improvement during training for the chosen loss function. Here, we use a binary cross entropy loss function, as is common with binary classification problems. With a small training set size of 1000, the learning curve for U-Net CNN is smooth and gradual but often becomes trapped in local optima: we enter stationarity at around 500 epochs, but our loss scores never approach those for larger training sets. When this happens, the U-Net CNN model has simply learned to always classify pixels as non-lipids, which scores reasonable well with our unbalanced data but is clearly non-optimal. The learning curves for training set sizes over 2000 are very similar as they approach zero loss at a similar rate. Interestingly, the CNN model trained using a large training set size of 5000 scored worse than other models built with smaller training sets (possibly due to chance or model overfitting, see Fig 11). For this binary segmentation task with these data, a training set size of 2000 images may be sufficient to produce the best trade-off of accuracy vs. computational speed.

tensors. This pattern of applying convolutional layers, upsampling, and concatenation continues until we result in a tensor with our original image 256x256 image dimensions. After a couple additional convolutional layers, our final output layer is implemented as a 2D convolutional layer with a sigmoid activation function as a continuous approximation to a binary step function. When using the trained model during classification, we apply a simple intensity threshhold cutoff (0.7) to post-process the continuous output values from the model to final binary classification output.

As with our prior evaluation of non-deep learning methods, we perform both a k-fold cross-validation with our U-Net CNN model as well as a traditional train-test split. For the train-test split case, we used different training set sizes between 1,000 and 5,000 images and halted at 1000 epochs. The learning curve, or rate of change of the loss function [132], was very rapid with the U-Net CNN (Fig 7). Notably, a training set size of 1000 images consistently resulted in a shallower learning curve that prematurely reached stationarity within localized optima at approximately 500 epochs. The classification output from models trained with only 1000 images never classified *any* inputs as LDs, and we determined that a non-augmented training set size of 1000 or less was too small for these inputs. However, CNN models trained

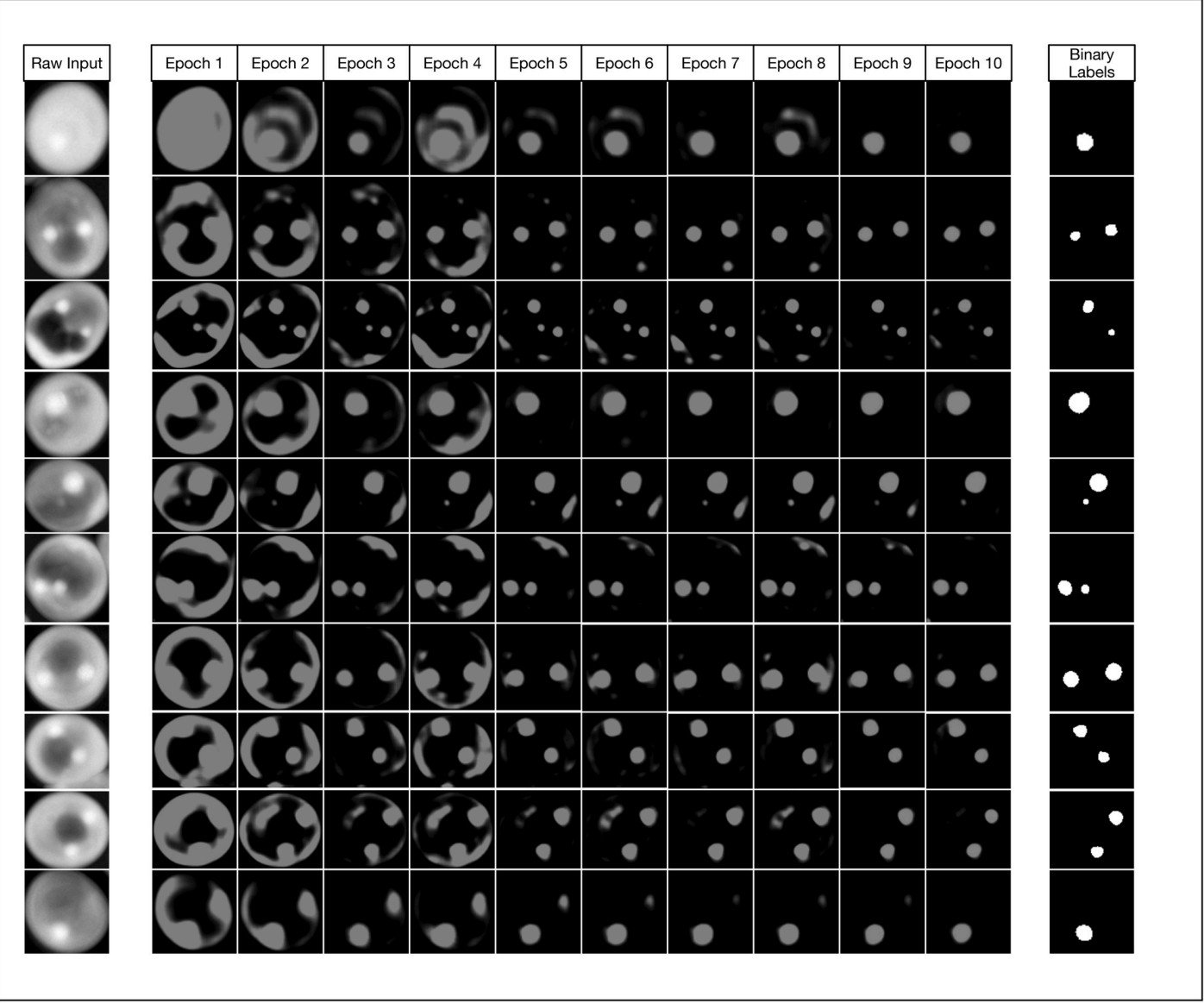

**Fig 8. Watching the U-Net convolutional network learn.** This figure demonstrates how the model noticeably improves very early during the initial 10 epochs of training. While we trained our models to 1000 epochs, we show that even by epoch 10 with a training set size of 5000 images, our model has already started approximating the true image segmentation (as shown in the rightmost column).

with 2,000 or more training images provided improved learning curves and improved effectiveness scores.

As an illustrative exercize to visualize the incremental improvements made by the U-Net CNN during the training phase, we instrumented our Python framework to save trained models after every epoch and perform a classification on test input (Fig 8). This qualitative visualization of the rate of improvement correlates to our quantitative learning curves (Fig 7), as there is exceptionally rapid improvement within just the first few epochs. By just the tenth epoch, it is visually obvious that in most cases the model is already closely approximating our binary labels for our test data. However, our learning curves indicate that our models continue to asymptotically improve across all 1,000 epochs when the training set contains 2,000 or more

images. We found that in most cases, relatively small improvements in binary cross entropy loss resulted in noticeable corresponding improvements in the model's effectiveness metrics.

## Results

We constructed our tests to provide a consistent framework to compare six different machine learning methods: random forest, gradient boosting with XGBoost, support vector machines, multilayer perceptron neural networks, linear discriminant analysis, and U-Net convolutional neural networks. With some notable exceptions, most methods performed similarly from a quantitative standpoint. We used the same stochastically grouped training and test sets for each method as shown earlier in Fig 2 and used the same or analogous hyperparameters across models wherever possible (Table 1). A diversity of quantitative metrics is required to fully understand a classifier's ability to correctly perform image segmentation [133]. We report results across seven standard metrics including accuracy, balanced accuracy, precision, recall, Sørensen-Dice ("Dice") coefficient [81–84] (*i.e.* F1 score), Jaccard distance, and Receiver Operating Characteristic (ROC) Area Under the Curve (ROC AUC) scores [134–136].

Our basic scoring functions are explicitly defined in terms of true positives (*TP*), true negatives (*TN*), false positives (*FP*) and false negatives (*FN*) when performing pixel-by-pixel comparison between a "true" labeled image and binary classified output from our machine learning models:

$$Accuracy = \frac{TP + TN}{TP + TN + FP + FN}$$

$$Balanced\ Accuracy = \frac{1}{2}\left(\frac{TP}{P} + \frac{TN}{N}\right)$$

$$Precision = \frac{TP}{TP + FP}$$

$$Recall = \frac{TP}{TP + FN}$$

$$Dice/F1 = \frac{2TP}{2TP + FP + FN}$$

$$Jaccard\ Index = \frac{TP}{TP + FP + FN}$$

While most non-deep methods were able to successfully train useful semantic segmentation classifiers with a small non-augmented training set of only 1,000 images, we found that the deep learning CNN method can occasionally become trapped in local optima during gradient descent optimization and produce a classifier that consistently produced blank output (i.e., no true *or* false positives). The lack of any positives can result in uninterpretable scores in some cases. Because of the high ratio of non-LD pixels in these images (*i.e.*, *unbalanced classes*), it is likely that the CNN partially optimized the classifier to always predict non-LD pixels, resulting in a reasonably low but far from globally optimal binary cross entropy loss during model training from which the gradient descent optimizer could not escape. We expect that this could be mitigated by augmenting our data during training to synthetically increase our pool of training images, modifying our learning rate, or preloading U-Net weights prior to training.

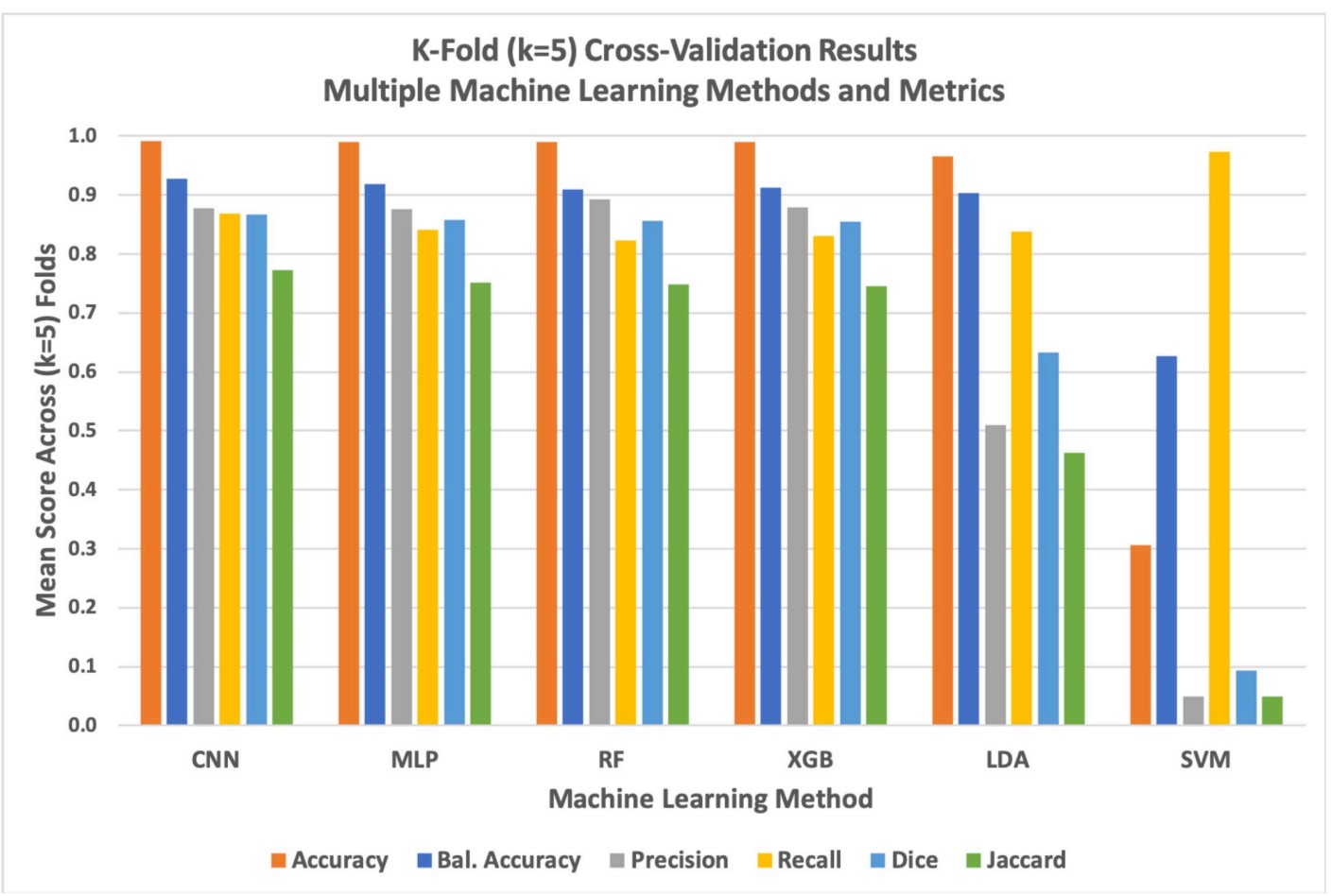

**Fig 9. Graph of k-fold cross-validation results for all six evaluated machine learning methods against six distinct performance metrics.** The deep learning model generally but marginally outperformed simpler methods while support vector machines (SVM) failed as evaluated against most quantitative metrics. We found that SVM tended to predict significant numbers of false positives, resulting in an anomalously high recall score.

We performed two distinct evaluations of our six machine learning methods: k-fold cross-validation and a separate train-test split against new test data. We performed the cross-validation using the 5,000-image training set as described previously in Fig 2. The k-fold cross-validation effectively divided the training set into $k = 5$ sets of 1,000 images each and iteratively trained using 4,000 images and tested against the 1,000-image hold-out set for each of the 5 folds. This statistical resampling approach is commonly used when evaluating machine learning models with limited training data and often results in less biased estimates of model effectiveness as compared to the simpler train-test split approach.

The deep learning U-Net CNN method quantitatively but narrowly outperformed other methods across most metrics during k-fold cross-validation (Fig 9 and Table 2). While the U-Net CNN achieved the highest average scores across 4 of the 6 metrics used during cross-validation, the overall quantitative performance of the Deep learning method was only marginally better than random forest, gradient boost, and simpler non-convolutional neural networks. Linear discriminant analysis had a mediocre performance, while support vector machines failed entirely against this image segmentation problem with these data.

In addition to k-fold cross-validation, we performed a simple test-train split of our original library of 7,000 images for two reasons: to produce comparative Receiver Operator

**Table 2. Comparing six machine learning methods using k-fold cross-validation.**

|  | CNN | MLP | RF | XGB | LDA | SVM |
|---|---|---|---|---|---|---|
| Accuracy | 0.9908 | 0.9900 | 0.9901 | 0.9898 | 0.9652 | 0.3058 |
| Balanced Accuracy | 0.9283 | 0.9184 | 0.9092 | 0.9129 | 0.9036 | 0.6271 |
| Precision | 0.8778 | 0.8761 | 0.8919 | 0.8795 | 0.5090 | 0.0489 |
| Recall | 0.8691 | 0.8412 | 0.8221 | 0.8301 | 0.8373 | 0.9733 |
| Dice | 0.8668 | 0.8577 | 0.8556 | 0.8541 | 0.6330 | 0.0931 |
| Jaccard | 0.7725 | 0.7509 | 0.7476 | 0.7453 | 0.4631 | 0.0488 |

To thoroughly validate our different machine learning methods against the problem of segmenting lipid droplets within QPI images of Y. lipolytica cells, we performed k-fold cross-validation with k = 5 folds. Scores for each metric are averaged across folds. Green within the table indicate the highest score for the respective metric. Here, we see that U-Net CNN narrowly outperformed the non-deep methods across most metrics. Generally, all evaluated methods performed similarly well with the notable exception of support vector machines which tended to produce an overabundance of false positives across all of our tests (note that the recall metric ignores false positives, leading SVM to score particularly well with that metric).

Characteristic (ROC) curve plots of all six evaluated machine learning methods (Fig 10) *and* to evaluate how training set size effects the accuracy and computational efficiency of the three most accurate methods. ROC curves are graphical plots of the True Positive Rate vs. the False Positive Rate and characterize the ability of a classifier to perform at different discrimination thresholds, independent of the class distribution within the training and test data.

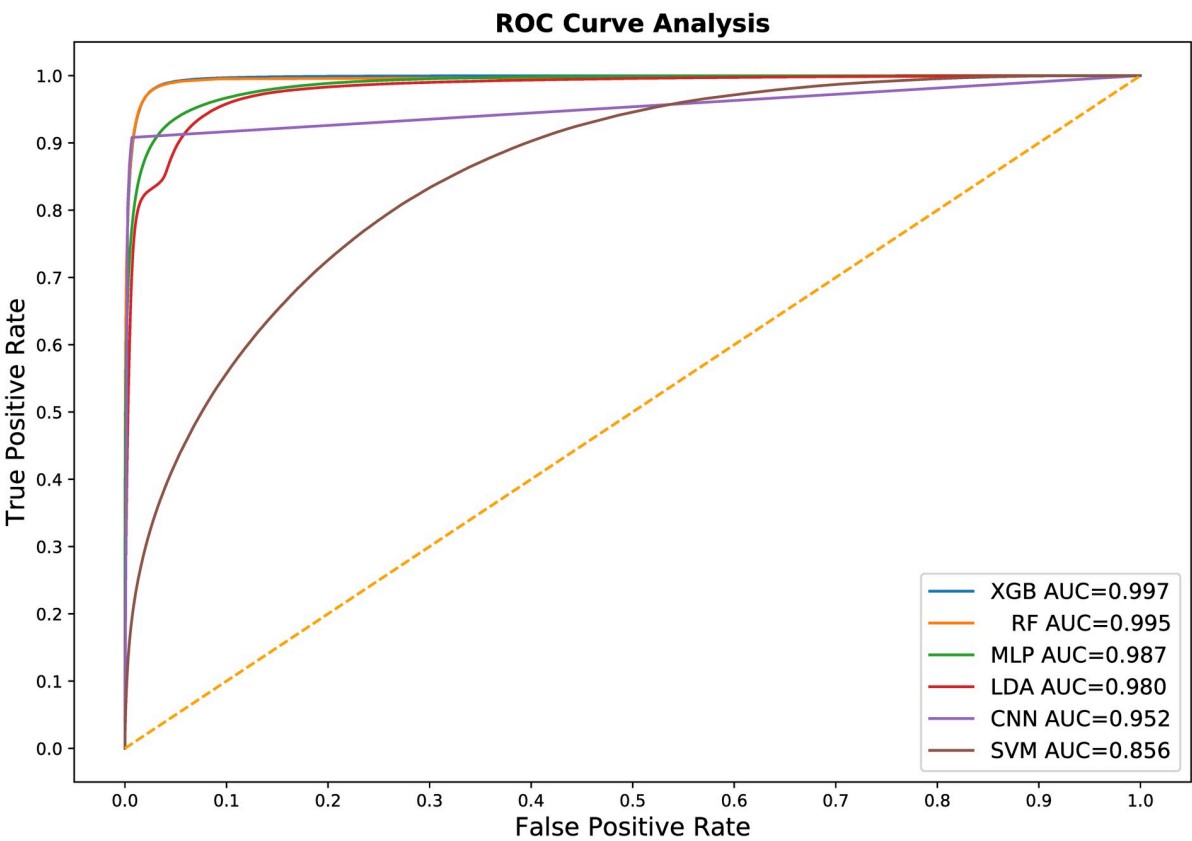

**Fig 10. The Receiver Operating Characteristic (ROC) curve for all six of our machine learning classifier models.** These curves and their corresponding Area Under the Curve (AUC) summary statistic generally match our k-fold cross validation results. In general, these curves indicate that most of the evaluated machine learning methods are comparable and highly effective in training a usable semantic segmentation classifier. The significant underperforming outlier is the support vector machines model.

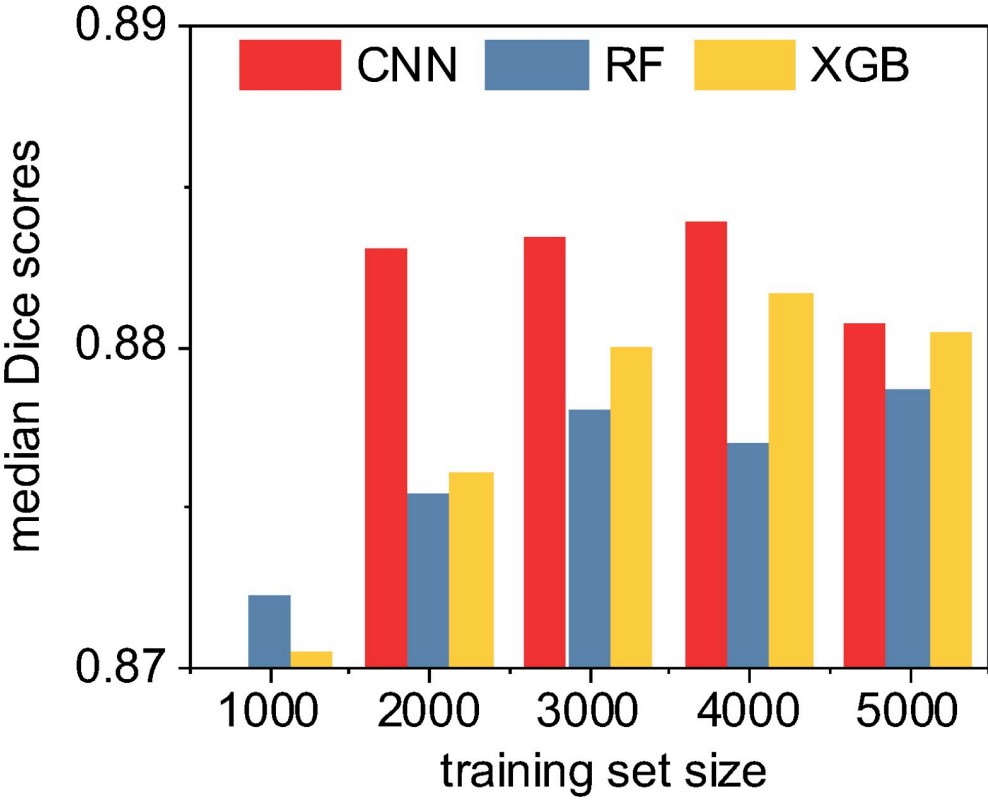

**Fig 11. A quantitative accuracy comparison of three machine learning methods as a function of training set size.** Here we compute the median Sørensen-Dice coefficient (i.e. "Dice" or F1 score) given by each method for each training set size. Note that the CNN was unable to consistently train an effective classifier with only 1000 non-augmented training images due to local optima traps; however, the deep learning CNN method otherwise consistently outperformed the ensemble classifiers. XGBoost generally outperformed random forest. Overall, in absolute terms as shown by the range on the y-axis, the practical quantitative differences between the methods are minimal.

In our last evaluation, we performed a limited set of tests specifically between the U-Net CNN and the high-performing ensemble methods (random forest and XGBoost) in order to characterize method performance for both accuracy and speed as a function of training test size. (Fig 11) In all tests, *median* Dice/F1 accuracy scores ranged in an exceptionally narrow band between 0.87 and 0.89. In general, XGBoost produced slightly more accurate classifiers than random forest and all methods improved as training set size increased, with the notable exception of the training set size of 5,000 images. With this relatively high number of training images, both U-Net CNN and XGBoost Dice scores declined slightly, possibly indicative of model overfitting or chance due to our specific randomized train-test split.

In addition to quantifying the relative accuracy of the different machine learning methods under different training set sizes, we also characterized the computational complexity and performance of these methods as a function of training set size. We demonstrated earlier in Fig 5 that the random forest and XGBoost methods were $O(n^2)$ with respect to the number of input pixels due to the expensive and required expensive internal data re-structuring prior model training. Using the Keras framework for our CNN implementation, we instead stream our training data though our model during training, resulting in a linear $O(n)$ complexity relative only to the number $n$ of input images (Fig 12). Despite the improved scalability of our CNN implementation, it takes considerably less wall-clock time to train non-convolutional methods

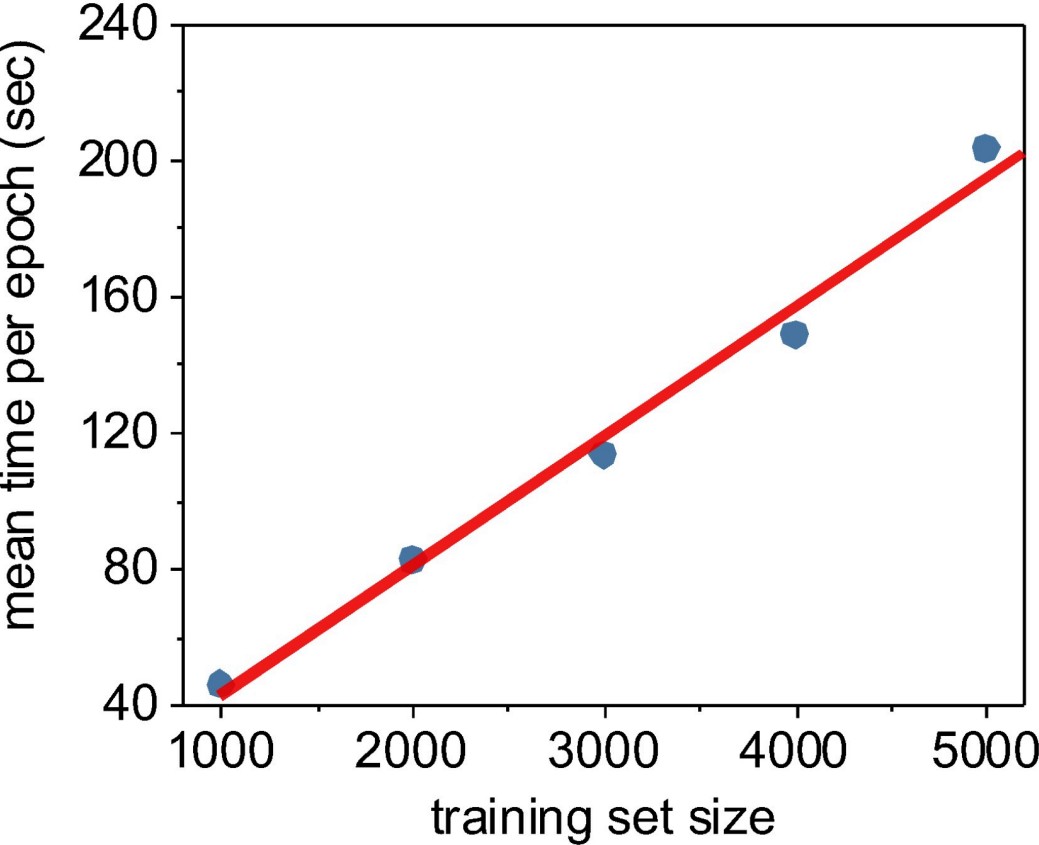

**Fig 12. The time required to train one epoch as a function of training set size.** For neural networks, an epoch is defined as one full pass of the training set through the model. Using the Keras Python framework, we stream our training data through our U-Net CNN model in batches, and thus the time required to train per epoch is essentially a linear function of the training set size. In this case, we trained using Keras/TensorFlow on a consumer-level GPU (Nvidia GeForce GTX 1080 Ti).

such as random forest or XGBoost due to the high *fixed complexity* of the U-Net architecture. In fact, we found that the only practical way to train our U-Net CNN in a reasonable time-frame was to use Keras [137] on a GPU-enabled TensorFlow backend [138].

While training was computationally expensive, all three of these tested supervised machine learning methods produced models which could perform full semantic segmentation on new images in a fraction of a second per image (Fig 13). Unsurprisingly, the U-Net CNN classifier implemented on an Nvidia 1080 Ti GPU accelerator significantly outperformed the conventional CPU-based methods with a median time of 15.4 msec per image. Our CPU-based tests were executed on a 28-core Dell PowerEdge M640 (2x 14 core 2.2 GHz Intel Xeon Gold 5210 CPUs with 512 GB RAM). XGBoost classifiers were also relatively fast, largely due to the shallow depth of the individual decision trees within the model, leading to fast classification of individual pixels. The slowest method tested was the U-Net CNN implemented on a conventional CPU, certainly due to the fixed high complexity of the U-Net CNN architecture. In all, each of these methods took less than half a second per full image segmentation, and all methods are practical candidates for integration into a complete high-throughput optical imaging and analysis workflow.

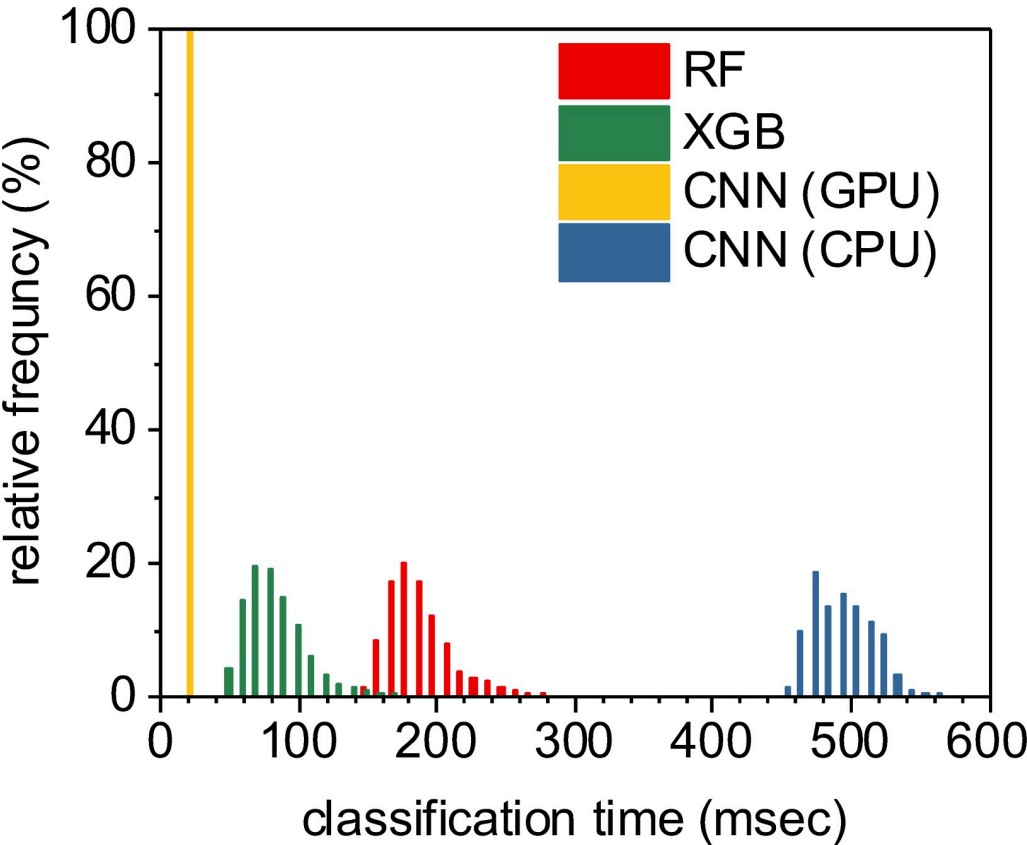

**Fig 13. Classifying all pixels within an image using a trained model is relatively fast across all machine learning methods used.** The fastest approach was the U-Net CNN executed on a Nvidia 1080 Ti GPU, with a median time per image segmentation of 15.4 msec. XGBoost classifiers were also relatively fast at a median rate of 76 msec per image segmentation. Random forest classifiers took a median time of 181 msec per image, while U-Net classifiers implemented on a CPU (instead of a GPU) took significantly longer at a median time of approximately 484 msec per image.

## Discussion

We tested six supervised machine learning semantic segmentation methods to classify pixels that correspond to LDs within images of *Y. lipolytica* cells. We assessed and compared classification accuracy and computational efficiency using both k-fold cross-validation and train-test split evaluations. For three of the best performing models, we also evaluated accuracy and computational time as a function of training set size. From a purely quantitative standpoint, most of these methods performed similarly with some noticeable differences. First, we found that a convolutional neural network using the popular U-Net architecture performed marginally better than simpler, non-deep learning methods, but required a sufficient training set size greater than 1,000 images. In reality, this is unlikely to be a significant problem for U-Net CNN as training set sizes of 2,000 and above were sufficiently rich to develop a robust classifier and synthetic data augmentation would almost certainly suffice to construct enough training data. Pre-loading known U-Net model weights would also likely help avoid local optima traps during training.

During k-fold cross validation, most of the evaluated machine learning methods scored similarly well across multiple evaluation metrics, while the deep learning CNN method scored only marginally better on the whole. Additionally, the similar ROC curves and AUC metrics

using a train-test split evaluation supported this assessment. From a purely quantitative assessment, either U-Net CNN, random forest, XGBoost, or multilayer perceptron neural network approach would likely work well. This indicates that supervised machine learning is an effective approach to the semantic segmentation of these QPI images.

We extended our evaluation by considering how model classification skill for our top scoring methods was affected by training set size. In general, accuracy increased as a function of training set size until our training set exceeded 5,000 images at which point there was a slight drop in classification accuracy scores for both the CNN and XGBoost methods. In general, adding more training data mitigates overfitting; it is possible that this case was a minor

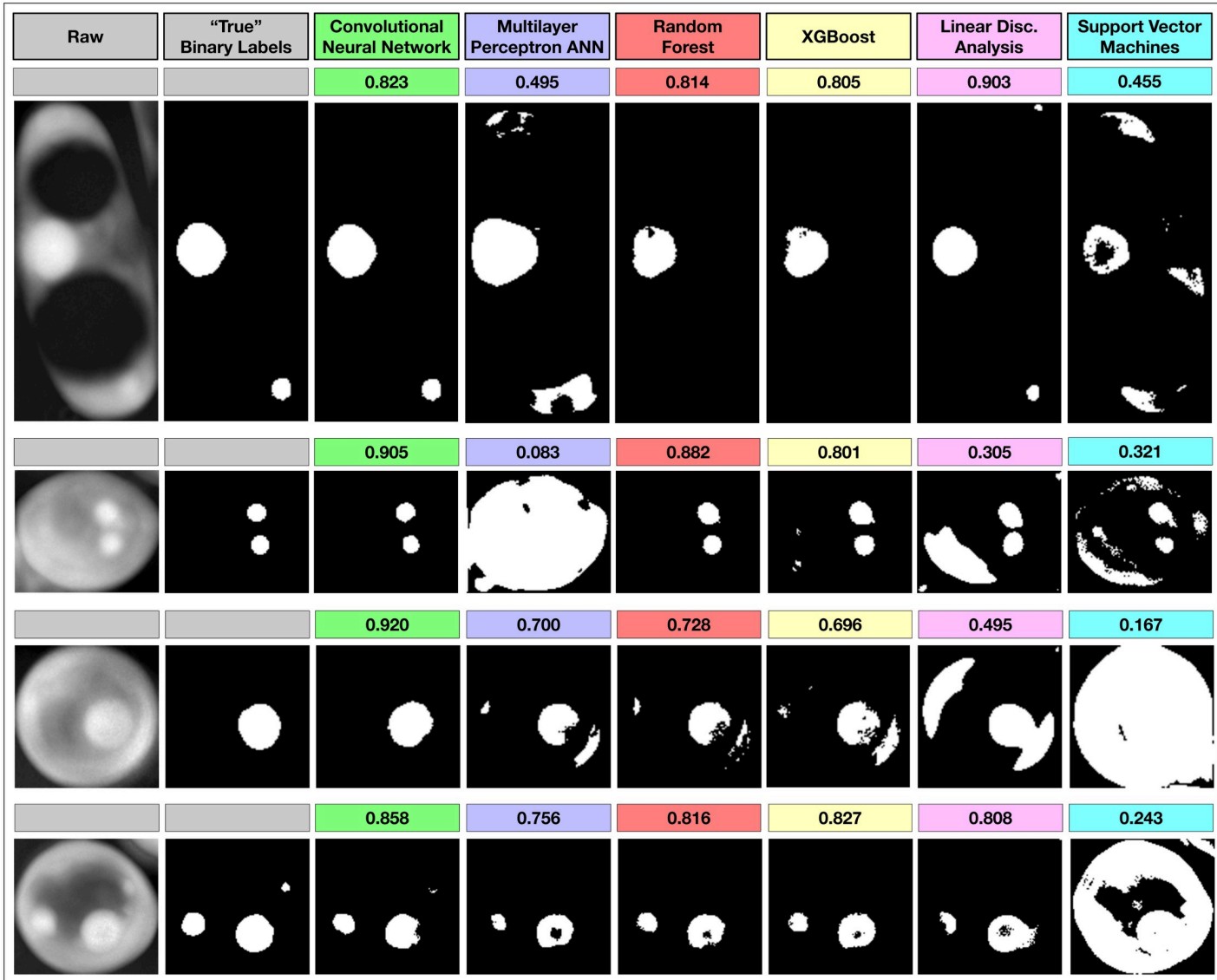

**Fig 14. Qualitative argument for the use of the U-Net CNN.** While non-deep learning methods can sometimes score similarly to deep learning methods such as the U-Net CNN, we found that deep learning methods produces smoother and more biologically interpretable segmentations in almost all cases. Computed Dice/F1 scores are shown above the images. These images are examples where other methods produced reasonably high scoring but qualitatively unrealistic or noisy classifications of lipid droplets. Among other reasons, this is likely because the U-Net CNN directly persists and integrates original 2D spatial information while building the segmentation map. With the other methods, this 2D information is only indirectly inferred in a lossy way via the particular image filters used during feature extraction.

stochastic anomaly and adding even more training data would result in modest increases in accuracy. It is also plausible that the specific additional 1,000 images did not have a sufficiently high signal-to-noise ratio to improve model generalization. However, in terms of quantitative Dice/F1 accuracy scores, all tested methods performed remarkably similarly for training set sizes of 2,000 images or more.

## Qualitative differences in biological interpretability

Importantly, despite the similar quantitative scores across methods, we found that the deep learning U-Net CNN method consistently provided more biologically interpretable and realistic image segmentations. We reviewed the output of the trained models for all methods to elucidate the *qualitative* differences between the methods (Fig 14). While these are merely demonstrative examples, it is clear that the U-Net CNN more often classifies LDs as round, smooth, continuous objects that are physically interpretable as realistic and monolithic organelles. Non-convolutional methods create similar output to U-Net CNN in most cases, but are more likely to produce noisy image segmentations with rough and irregular edges, spurious single-pixel misclassifications, or holes and openings within otherwise complete and round droplets. In addition to the higher quantitative scores across multiple metrics, this *qualitative* advantage for the U-Net CNN is important to consider when deciding which method to adopt.

Because of the fixed complexity of the U-Net model, we found that random forest and XGBoost were much faster to train despite their greater asymptotic computational complexity as a function of the number of training images. In fact, training a U-Net CNN for 1,000 epochs using multicore CPUs was impractically slow for larger training sets. However, using GPU support with our Keras/TensorFlow implementation led to 40X speedups in both training and classifications time. Thus, using a U-Net CNN executed on modern GPUs to classify LDs produces the fastest, most accurate and interpretable segmentations in QPI images.

## Supporting information

**S1 Fig. The supervised machine learning steps used to train a non-deep, non-convolutional classifier to label pixels corresponding to subcellular lipid droplets.**
(PNG)

**S1 File. Feature importance is easily determined by evaluating the average relative position of each feature across all decision trees in either the random forest or XGBoost methods.** Features closer to the root of the trees are more important to overall classification decision. This spreadsheet describes the relative importance of the 80 extracted features from a trained random forest model, describes the image filtering functions use for feature extraction including their parameterization, and defines which Python libraries were used. Model transparency and interpretability is a key advantage of decision-tree based methods such as random forest or XGBoost. Other methods, such as neural networks, are often impossible to interpret or understand.
(XLSX)

## Author Contributions

**Conceptualization:** Luke Sheneman, Andreas E. Vasdekis.

**Data curation:** Luke Sheneman, Gregory Stephanopoulos.

**Funding acquisition:** Andreas E. Vasdekis.

**Investigation:** Luke Sheneman.

**Methodology:** Luke Sheneman, Gregory Stephanopoulos, Andreas E. Vasdekis.

**Project administration:** Andreas E. Vasdekis.

**Resources:** Luke Sheneman, Gregory Stephanopoulos.

**Software:** Luke Sheneman.

**Validation:** Andreas E. Vasdekis.

**Visualization:** Luke Sheneman.

**Writing – original draft:** Luke Sheneman.

**Writing – review & editing:** Andreas E. Vasdekis.

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
