## [Decision Letter · Decision Letter 0]

27 Oct 2020

PONE-D-20-23589

Deep learning classification of lipid droplets in quantitative phase images

PLOS ONE

Dear Dr. Sheneman,

Thank you for submitting your manuscript to PLOS ONE. After careful consideration, we feel that it has merit but does not fully meet PLOS ONE’s publication criteria as it currently stands. Therefore, we invite you to submit a revised version of the manuscript that addresses the points raised during the review process.

Especially, the reviewers suggest to show the performance of the proposed method under varying illumination conditionals or magnification rate and also strongly recommend sharing the data on public repository as required by PLOS One too to improve reproducibility of the work.

We look forward to receiving your revised manuscript.

Kind regards,

Jianjun Hu, Ph.D

Academic Editor

PLOS ONE

Journal Requirements:

Reviewers' comments:

Reviewer's Responses to Questions

**Comments to the Author**

1. Is the manuscript technically sound, and do the data support the conclusions?

Reviewer #1: Yes

Reviewer #2: Yes

2. Has the statistical analysis been performed appropriately and rigorously? 

Reviewer #1: Yes

Reviewer #2: Yes

3. Have the authors made all data underlying the findings in their manuscript fully available?

Reviewer #1: No

Reviewer #2: No

4. Is the manuscript presented in an intelligible fashion and written in standard English?

Reviewer #1: Yes

Reviewer #2: Yes

5. Review Comments to the Author

Reviewer #1: Reviewer’s comment

The authors reported the application of deep-learning in recognizing and localizing lipid droplets (LDs) directly in quantitative-phase images of yeast cells without staining. The purpose is to improve the discriminatory power and the specificity of LD localization in single living cells by Quantitative Phase Image (QPI) analysis. It also aims to address the deficiency of the existing method of coupled deconvolution- and correlation-based post-processing schemes, which is computationally costly thus not suitable for automated high-throughput QPI image processing. To address the problem, the authors applied the Deep Neural Networks, the Random Forest and the Gradient Boosting algorithms to the Binary classification of image pixels for image segmentation and LD localization. They performed experiments on QPI image datasets and demonstrated that the Deep Neural Network model outperformed the other two methods with significantly improved accuracy in LD classification and reduced computation time.

The manuscript was well written and the figure quality is good. Here are a number of questions to be made clear,

1. Over the past decades, numerous computer algorithms have been developed for solving image segmentation and object recognition problems (Pal et al. 1993, Zhao et al. 2019). It has been reported that the standard decision tree algorithm and its extensions performed not so well when compared to artificial neural network models on image recognition tasks. As the authors were training binary classifier models at single pixel level on the QPI image dataset, it might be reasonable to know if the simpler neural network models, such as Support Vector Machines (SVMs) or Fisher’s Discriminant Analysis (FDA), can simply perform equally well or even better than the Deep Neural Network model for LD identification?

2. Will the developed method also be applicable to LD pattern recognition under high noise background when cell debris, cell clump or other small particles are present in the imaging field of view? Can the method perform well under varied conditions of illumination and scale/magnification rate of microscopic imaging?

3. The section of introduction on Decision tree and ensemble leaning algorithms is redundant and can be moved to Supplementary method. The description of these algorithms can be found in the textbook of Machine Learning course. Instead, a more detailed introduction on the training process of U-net, including the pre-processing steps of the QPI image data by the authors, shall be presented mathematically by standard.

4. A Receiver Operating Characteristic Curve plot with calculated Area Under Curve (AUC) score will be informative for taking a fair comparison on the performance of the algorithms used for LD localization.

5. A cartoon diagram similar to Figure 1 in paper Reference 53 will be help people to better understand the experiment

and data processing workflow of this study.

Finally, I recommend that the paper should be accepted for publication after all these questions are cleared.

Reference

Pal, N. R., & Pal, S. K. (1993). A review on image segmentation techniques. Pattern recognition, 26(9), 1277-1294.

Zhao, Z. Q., Zheng, P., Xu, S. T., & Wu, X. (2019). Object detection with deep learning: A review. IEEE transactions on neural networks and learning systems, 30(11), 3212-3232.

Reviewer #2: Deep learning classification of lipid droplets in quantitative phase images

In this paper, the author explored using machine learning method to identify lipid droplets in QPIs. Though I’m not familiar with lipid droplets’ functions in biology, according to author’s introduction, it’s a very critical topic in bio molecular.

In this work, the author prepared and labeled training dataset, a group of images with lipid droplets in it. Author explored two major methods on this dataset: decision tree and CNN (U-NET). According to author’s experiment, CNN outperformed decision tree.

Overall, it’s a very interesting article. If the dataset is public available and author could address below comments, I think it’s a solid work to publish.

Major comments:

In this work, the author only conducted experiments on a one-time split train/test samples. A more robust and standard results could be obtained by conducting a N-fold cross-validation experiment. This will give more convincing results. Thus, I’m requesting the author to conduct a 5- or 10-fold cross-validation experiment on all 3 models.

Is the dataset public available? I was unable to find a link to it in this article. For CV work, I believe the dataset is critical to reproduce and verify the work.

“The library consists of two Y. lipolytica strains, Po1g and MTYL038..”

What’s the difference of these two strains in terms of appearing in pictures? In the training data labeling, did author label them differently or mark both of them just as positive pixels?

Minor comments:

Can user give the number of trainable parameters in this U-net?

“We sequentially preprocess every image in our training set in this way and flatten our internal 3D representation of the training set data with extracted per-pixel features into a long 2D array”

Can you add the shape size of each sample into the article? I think it should be [m*n, 80]?

Supporting Information Figure 2 is actually a table. Why not prepare it as an Excel file such that other researchers can easily query? Right now, the pic resolution is very low and hard to check. Also, maybe add another column in which author can list Python package you are using for each feature.

In CNN training, does author use any data augmentation approaches? It’s a very commonly used method in CV. If author used it, may need to mention it explicitly.

“As such, QPI is not compatible with automated high-throughput image processing in LD localization LDs, with the exception of coupled deconvolution- and correlation-based post-processing schemes, albeit at increased computational resource requirements and error-rates [53].”

[…in LD localization LDs…], not sure what does it mean.

“These stored machine learning models can also be inspected to report metadata such as relative feature importance (Sup. Fig. 2).”

Which approach did author use to calculate the feature importance in this Sup. Fig.?

6. PLOS authors have the option to publish the peer review history of their article (what does this mean?). If published, this will include your full peer review and any attached files.

Reviewer #1: No

Reviewer #2: No

---

## [Author Response · Author response to Decision Letter 0]

9 Jan 2021

Reviewer 1 

Comment 1: Over the past decades, numerous computer algorithms have been developed for solving image segmentation and object recognition problems (Pal et al. 1993, Zhao et al. 2019). It has been reported that the standard decision tree algorithm and its extensions performed not so well when compared to artificial neural network models on image recognition tasks. As the authors were training binary classifier models at single pixel level on the QPI image dataset, it might be reasonable to know if the simpler neural network models, such as Support Vector Machines (SVMs) or Fisher’s Discriminant Analysis (FDA), can simply perform equally well or even better than the Deep Neural Network model for LD identification?

Response: We appreciate this feedback and felt that this is a point worth exploring. In addition to the original three methods as described in the initial manuscript submission, we have now expanded our analysis to include the use of three additional “simpler” machine learning methods: 1) Support Vector Machines (SVMs), 2) linear discriminant analysis (LDA), and 3) simple multilayer perceptron (MLP) neural networks. In addition, we further enhanced our methods by using wider variety of classifier performance metrics including accuracy, balanced accuracy, precision, recall, Jaccard distance, and more. 

We found that MLP with 2 hidden layers (50 and 25 nodes each) performed well during k-fold cross-validation but SVMs did not generally perform well and tended to result in a significant false positive rate. LDA classifiers performed reasonably well but were not as accurate as most of the evaluated machine learning methods on this specific segmentation problem. 

Our revised and extended analysis of additional machine learning methods is distributed throughout this revised manuscript. However, the relative performance of the various methods is best described in the new k-fold cross-validation results shown in Figure 9, Figure 10, and Table 2.

Comment 2: Will the developed method also be applicable to LD pattern recognition under high noise background when cell debris, cell clump or other small particles are present in the imaging field of view? Can the method perform well under varied conditions of illumination and scale/magnification rate of microscopic imaging?

Response: This is an important point that we did not adequately address in the original manuscript. Overall, QPI relies on the detection of the optical phase (rather than intensity) of the transmitted light through the sample. As such, any illumination/background inhomogeneities are eradicated from the image. For the same reason, any sample-based background heterogeneities (e.g., cell debris) can be easily detected and removed during cell segmentation by (e.g.,) size thresholding, without affecting the process of image formation. To this end, we have previously described these QPI advantages here (also reference [10] in the original and revised versions of the manuscript), under various levels of magnification. To address this comment, we performed the following modifications in the manuscript:

From (page 3, line 57): “Further, QPI enables high-contrast imaging between cells and their background, which has found applications in localizing the contour of individual cells (i.e., performing cell segmentation) without any computationally intensive approaches [10, 11].”

To (page 4, line 57): “Further, by detecting optical phase rather than intensity, QPI enables high-contrast cell imaging with minimal background heterogeneity, thus, enabling the contour localization of individual cells (i.e., performing cell segmentation) without any computationally intensive approaches [10, 11].”

Comment 3: The section of introduction on Decision tree and ensemble leaning algorithms is redundant and can be moved to Supplementary method. The description of these algorithms can be found in the textbook of Machine Learning course. Instead, a more detailed introduction on the training process of U-net, including the pre-processing steps of the QPI image data by the authors, shall be presented mathematically by standard.

Response: As recommended, the sections detailing the specifics of decision tree ensemble methods has been removed from the main body of the manuscript. With respect to preprocessing QPI image data prior to use in training the U-Net CNN: there is relatively minimal preprocessing required. Specifically, we need to do only two things to preprocess the images: 1) convert the 32-bit tiff images to a normalized 8-bit grayscale representation, 2) evenly pad the images with negative padding so that all images are a consistent 256x256 size.

To (page 14, line 281): “This is trivially done by computing the amount of border to evenly add to the top, bottom, and sides of the raw source image to result in a 256x256 image subsequently calling the ImageOps.expand() function within the Pillow (PIL) Python library.”

Comment 4: A Receiver Operating Characteristic Curve plot with calculated Area Under Curve (AUC) score will be informative for taking a fair comparison on the performance of the algorithms used for LD localization.

Response: This is an excellent point and we have added a Receiver Operating Characteristic (ROC) curve plot for all 6 machine learning methods, including an Area Under Curve (AUC) score. This has been added to the revised manuscript as Figure 10:

Figure 10: The Receiver Operating Characteristic (ROC) curve for all six of our machine learning classifier models. These curves and their corresponding Area Under the Curve (AUC) summary statistic generally match our k-fold cross validation results. In general, these curves indicate that most of the evaluated machine learning methods are comparable and highly effective in training a usable semantic segmentation classifier. The significant underperforming outlier is the support vector machines (SVM) model.

Comment 5: A cartoon diagram similar to Figure 1 in paper Reference 53 will be help people to better understand the experiment and data processing workflow of this study. 

Response: We agree and added a cartoon diagram in Figure 1 and accordingly revised its caption as follows:

Figure 1: (a) Cartoon diagram of the QPI image formation through the detection of the optical phase-delay of light transmitted through the cell cytosol (ΔΦcyt) and cytosolic LDs (ΔΦLDs > ΔΦcyt due to the innate refractive index differences) at a constant background (ΔΦbg = 0). (b) Representative Yarrowia lipolytica QPI images (in blue) overlaid with binary masks (magenta) acquired by direct thresholding (left column) and deep learning (middle column). The decreased discriminatory power of direct thresholding and the increased precision of deep learning become evident upon comparison with the ground truth (right column).

 

Reviewer 2

Comment 1: In this work, the author only conducted experiments on a one-time split train/test samples. A more robust and standard results could be obtained by conducting a N-fold cross-validation experiment. This will give more convincing results. Thus, I’m requesting the author to conduct a 5- or 10-fold cross-validation experiment on all 3 models.

Response: This is very valid point, and we have addressed it by performing a 5-fold cross validation for all 6 models. Further, in our cross-validation results, we extend our analysis beyond using Dice score and report 6 distinct accuracy metrics to provide a more nuanced and holistic understanding of the relative performance of these classifiers. The k-fold cross validation results are summarized in the revised manuscript in Figure 9 and Table 2:

Figure 9: Graph of k-fold cross-validation results for all six evaluated machine learning methods against six distinct performance metrics. The deep learning model generally but barely outperformed simpler methods while support vector machines (SVM) failed as evaluated against most quantitative metrics. We found that SVM tended to predict significant numbers of false positives, resulting in an anomalously high recall score.

Table 2: Comparing six machine learning methods using k-fold cross-validation. To thoroughly validate our different machine learning methods against the problem of segmenting lipid droplets within QPI images of Y. lipolytica cells, we performed k-fold cross-validation with k=5 folds. Scores for each metric are averaged across folds. Green within the table indicate the highest score for the respective metric. Here, we see that U-Net CNN narrowly outperformed the non-deep methods across most metrics. Generally, all evaluated methods performed similarly well with the notable exception of Support Vector Machines which tended to produce an overabundance of false positives across all of our tests (note that the recall metric ignores false positives, leading SVM to score particularly well with that metric).

 CNN MLP RF XGB LDA SVM

Accuracy 0.9908 0.9900 0.9901 0.9898 0.9652 0.3058

Balanced Accuracy 0.9283 0.9184 0.9092 0.9129 0.9036 0.6271

Precision 0.8778 0.8761 0.8919 0.8795 0.5090 0.0489

Recall 0.8691 0.8412 0.8221 0.8301 0.8373 0.9733

Dice 0.8668 0.8577 0.8556 0.8541 0.6330 0.0931

Jaccard 0.7725 0.7509 0.7476 0.7453 0.4631 0.0488

Comment 2: Is the dataset public available? I was unable to find a link to it in this article. For CV work, I believe the dataset is critical to reproduce and verify the work.

Response: Our original intent was to make the image dataset available upon publication. To address this Reviewer concern, we have made the source code and dataset available as now described at the end of the revised manuscript:

Data and Source Code:

All source code used in training, testing, and analysis is licensed under the MIT open-source license and can be found at: https://github.com/sheneman/deep_lipid

All of the data related to this research is publicly available and licensed under the Creative Commons Attribution Non-Commercial Share-Alike license. The dataset is available at: https://doi.org/10.7923/3d0d-yb04.

Comment 3: “The library consists of two Y. lipolytica strains, Po1g and MTYL038.” What’s the difference of these two strains in terms of appearing in pictures? In the training data labeling, did author label them differently or mark both of them just as positive pixels?

Response: The Reviewer correctly requests for clarification on the two strains. Briefly, the two strains exhibit minimal differences in terms of lipid content (only 1.5% greater lipid content in Po1g) and, thus, the resulting images. Further, the strains were not labelled differently in neither the training nor the classification steps. To clarify this key point, we performed the following modification:

From (page 6, line 110): “The library consists of two Y. lipolytica strains, Po1g and MTYL038 [54] grown under selected growth conditions that yield lipid content greater than 1.5% per weight. These conditions pertain to time points greater than 28 hours in a defined YSM medium with a carbon-to-nitrogen ratio (C/N) equal to 150.”

To (page 6, line 116): “The library consists of two Y. lipolytica strains, Po1g and MTYL038 [54], with the former being auxotrophic for leucine (Leu-) and yielding 1.5% greater lipid content per weight at identical growth conditions (i.e., 100 hrs in a carbon-to-nitrogen ratio (C/N) equal to 150) [53]. The two strains exhibited similar images at these conditions and were not labelled specifically during training and classification. To additionally include images with both low (smaller LDs) and high (larger LDs) lipid content, MTYL038 images were collected at different timepoints (28 hr, 52 hr, 76 hr, and 124 hr) in the same medium (C/N=150). In this context, cells exhibiting both low (early time points) and high (late time points) lipid content were attained [53]. Similarly, the different time points for MTYL038 were not labelled specifically during training and classification. .”

“Minor” Comment 4: Can user give the number of trainable parameters in this U-net?

Response: We have added this detail in the revised manuscript.

To (page 15, line 297): “In total, our U-Net CNN implementation has 31,031,685 trainable parameters.”

“Minor” Comment 5: “We sequentially preprocess every image in our training set in this way and flatten our internal 3D representation of the training set data with extracted per-pixel features into a long 2D array”

Can you add the shape size of each sample into the article? I think it should be [m*n, 80]?

Response: The Reviewer is completely correct. For each image of width w and height h, the flattened shape is [w*h,80]. For an entire training set of s images, the flattened shape of the long array is [∑_(i=1)^s▒〖w_i×h〗_i, 80]. We have added this detail to the revised manuscript for clarity:

To (page 11, line 223): “In total, we preprocess every ith 2D input image of width wi and height hi and a depth of k=80 features into a 3D array of dimension (wi,hi,k=80).”

To (page 11, line 226): “We sequentially preprocess every image in our training set of s images in this way and flatten our internal 3D representation of the training set data with k=80 extracted per-pixel features into a single long 2D array with ∑_(i=1)^s▒〖w_i×h〗_i rows and k=80 columns (Figure 3c)”

“Minor” Comment 6: Supporting Information Figure 2 is actually a table. Why not prepare it as an Excel file such that other researchers can easily query? Right now, the pic resolution is very low and hard to check. Also, maybe add another column in which author can list Python package you are using for each feature.

Response: We have converted Supporting Information Figure 2 to an Excel file instead of an image and have added a column indicating which Python library is used for each of the feature extraction filters.

“Minor” Comment 7: In CNN training, does author use any data augmentation approaches? It’s a very commonly used method in CV. If author used it, may need to mention it explicitly.

Response: We did not perform any data augmentation in any of the reported analyses, as we found that the size of the dataset was adequate to perform the reported analyses. To clarify this important point, we added the following sentence:

To (page 7, line 125): “We found that our curated image library is sufficiently large to train robust classifiers without the need for data augmentation of the acquired data.”

“Minor” Comment 8: “As such, QPI is not compatible with automated high-throughput image processing in LD localization LDs, with the exception of coupled deconvolution- and correlation-based post-processing schemes, albeit at increased computational resource requirements and error-rates [53].” […in LD localization LDs…], not sure what does it mean.

Response: We thank the Reviewer for bringing to our attention this typographical error. We have corrected it as follows:

To (page 5, line 88): “As such, QPI is not compatible with automated high-throughput image processing in LD localization, with the exception of coupled deconvolution.”

“Minor” Comment 9: “These stored machine learning models can also be inspected to report metadata such as relative feature importance (Sup. Fig. 2).” Which approach did author use to calculate the feature importance in this Sup. Fig.?

Response: We reported the feature importance from the Random Forest (RF) classifier using the intrinsic capability within the Scikit-Learn Python library to track and expose impurity-based feature importance of the forest within the trained model (i.e. we interrogated rf_classifier.feature_importances_ within our trained Random Forest model). 

This will perhaps be clearer if one reviewed our source code. The mathematics behind this is described in more detail here:

https://towardsdatascience.com/the-mathematics-of-decision-trees-random-forest-and-feature-importance-in-scikit-learn-and-spark-f2861df67e3

---

## [Decision Letter · Decision Letter 1]

15 Mar 2021

Deep learning classification of lipid droplets in quantitative phase images

PONE-D-20-23589R1

Dear Dr. Sheneman,

We’re pleased to inform you that your manuscript has been judged scientifically suitable for publication and will be formally accepted for publication once it meets all outstanding technical requirements.

Kind regards,

Yuchen Qiu, Ph.D.

Academic Editor

PLOS ONE

Additional Editor Comments (optional):

Reviewers' comments:

Reviewer's Responses to Questions

**Comments to the Author**

1. If the authors have adequately addressed your comments raised in a previous round of review and you feel that this manuscript is now acceptable for publication, you may indicate that here to bypass the “Comments to the Author” section, enter your conflict of interest statement in the “Confidential to Editor” section, and submit your "Accept" recommendation.

Reviewer #1: All comments have been addressed

Reviewer #2: All comments have been addressed

2. Is the manuscript technically sound, and do the data support the conclusions?

Reviewer #1: Yes

Reviewer #2: Yes

3. Has the statistical analysis been performed appropriately and rigorously? 

Reviewer #1: Yes

Reviewer #2: Yes

4. Have the authors made all data underlying the findings in their manuscript fully available?

Reviewer #1: Yes

Reviewer #2: Yes

5. Is the manuscript presented in an intelligible fashion and written in standard English?

Reviewer #1: Yes

Reviewer #2: Yes

6. Review Comments to the Author

Reviewer #1: The authors have fully addressed all the questions. The paper is sutiable to be accepted for publication.

Reviewer #2: In this revised version, the author thoroughly addressed all major comments in our previous comments. Thanks the author for carefully responding. Also, they give the source of the data and model they used in this research, which is very convincing.

I don't have more requests and I suggest editor to accept this research paper.

7. PLOS authors have the option to publish the peer review history of their article (what does this mean?). If published, this will include your full peer review and any attached files.

Reviewer #1: No

Reviewer #2: No

---

## [Editor Report · Acceptance letter]

19 Mar 2021

PONE-D-20-23589R1 

Deep learning classification of lipid droplets in quantitative phase images 

Dear Dr. Sheneman:

I'm pleased to inform you that your manuscript has been deemed suitable for publication in PLOS ONE. Congratulations! Your manuscript is now with our production department. 

Kind regards, 

on behalf of

Dr. Yuchen Qiu 

Academic Editor

PLOS ONE